# Homokaryotic High-Quality Genome Assembly of Medicinal Fungi *Wolfiporia hoelen* Reveals Auto-Regulation and High-Temperature Adaption of Probable Two-Speed Genome

**DOI:** 10.3390/ijms231810484

**Published:** 2022-09-10

**Authors:** Shoujian Li, Guoliang Meng, Caihong Dong

**Affiliations:** 1State Key Laboratory of Mycology, Institute of Microbiology, Chinese Academy of Sciences, Beijing 100101, China; 2University of Chinese Academy of Sciences, Beijing 100049, China

**Keywords:** ploidy, transposon, accessory chromosome, gene family expansion, methylation, mating recognition

## Abstract

Sclerotia of *Wolfiporia hoelen* are one of the most important traditional Chinese medicines and are commonly used in China, Japan, Korea, and other Asian countries. In the present study, we presented the first high-quality homokaryotic genome of *W. hoelen* with 14 chromosomes which was evaluated with assembly index, telomere position detection, and whole-genome collinearity. A 64.44 Mb genome was assembled with a Contig N50 length of 3.76 Mb. The imbalanced distribution of transposons and chromosome characters revealed the probable two-speed genome of *W. hoelen*. High consistency between methylation and transposon conserved the genome stability. The expansion of the gene family about signal transduction and nutritional transport has intimate relationships with sclerotial formation. Up-regulation of expression for distinctive decomposition enzymes, ROS clearance genes, biosynthesis of unsaturated fatty acids, and change of the cell wall components maintained high-speed growth of mycelia that may be the high-temperature adaption strategy of *W. hoelen*. Further, the analysis of mating-control genes demonstrated that *HD3* probably had no function on mating recognition, with the HD protein in a distant genetic with known species. Overall, the high-quality genome of *W. hoelen* provided crucial information for genome structure and stability, high-temperature adaption, and sexual and asexual process.

## 1. Introduction

Fungi are filamentous living organisms that rot easily, so there were few related and constant records. However, there are a few fungi that have been used for thousands of years, such as traditional Chinese fungal medicines Lingzhi (*Ganoderma* spp.), Fuling (*Wolfiporia hoelen*), and Sanghuang (*Sanghuangporus* sp., *Phellinus* sp., *Inonotus* sp., *Fomitiporia* sp.) [1,2,3]. According to Wu et al. and Stalpers et al., *Wolfiporia hoelen* (Fr.) Y.-C. Dai and V. Papp (syn. *Poria cocos*, *Wolfiporia cocos*) was the scientific name of Chinese “Fuling”, which grows on the root of pines all year round and is widely distributed in East Asia. “Fuling” was considered to be the same species as the American tuckahoe, until it was recently found to differ from the American species *W. cocos*, according to phylogenetic analyses and morphological examination, and revised to *W. hoelen* [4,5].

*W. hoelen* was bracket fungus which can form fruiting bodies with whitish resupinate poroids and asexual structure sclerotia. The fruiting bodies were first discovered by Wolf [6], with induction conditions conducted frequently in the past 10 years [7,8,9]. The sclerotia were the edible and medicinal structure of *W. hoelen*, which was cultivated in China for 1500 years, and widely cultivated in many provinces of China and Korea [10,11]. In traditional Chinese medicine, Fuling was widely involved in Chinese medicinal formulas, so there was a common saying in China, that there would be nine Chinese medicinal formulas containing Fuling among ten [12]. Various active ingredients, mainly including triterpenoids and polysaccharides, and multiple pharmacological activities of Fuling have been demonstrated in modern medicine [13,14,15,16].

To reveal the mechanism of sclerotial formation, wood degradation, and biosynthesis of medicinal components polysaccharides and triterpenoids, genome and transcriptome sequencing were conducted [17,18,19,20]. The genome of single spore isolates MD-104 SS10 (American strain *W. cocos*) was sequenced using a hybrid whole-genome shotgun approach with a combination of ABI3730 (fosmids) (Applied Biosystems, Foster City, CA, USA), 454-Titanium (454 Life Sciences, Branford, CT, USA), and Illumina GAII (Illumina Inc., San Diego, CA, USA) sequencing platforms, which had previously been used as the reference genome [21,22]. However, the American species has been demonstrated to differ from Chinese *W. hoelen* [4,5]. Luo et al. [23] published a Chinese Fuling genome (strain CGMCC5.78), which was sequenced using the Illumina Hiseq 2000 platform through the construction of the fosmid library, but the strain used should be heterozygous and consist of two sets of heterogeneous nuclear genomes since there was no work on the identification of the homokaryotic strain. The genes involved in sclerotia formation, fruiting, mating, secondary metabolism, and CAZymes were all annotated, with the differentially expressed genes between mycelia and sclerotia analyzed. Cao et al. [24] further published a chromosome-level genome with an ambiguous karyotype, probable incorrect chromosome number, and of low quality. The gene family expansion and extraction, CAZymes and CYP450, were further analyzed. In our previous research, the homokaryotic SSIs (single spore isolates) were collected and verified through heterozygous ratio estimation of the genome [9], which provided the possibility of the high-quality genome assembly.

Previous results remind us of the genomic complexity of *W. hoelen*, especially the possible high ratio of transposons [24]. A two-speed genome model has emerged in filamentous plant pathogens because the effector genes, encoding virulence effectors, tend to be associated with compartments enriched in repetitive sequences and transposable elements [25]. The variability of homokaryotic strains discovered in our previous research [26] may be caused by genetic crossover which can be understood easily, but also may imply the unusual genome of *W. hoelen*. Being different to the majority of edible and medicinal mushrooms, *W. hoelen* is cultivated in fields and has to endure high temperatures (>30 °C) every day during torrid summers. In addition, the migration route of cultivation districts is clear from the north to south of China, similar to the migration from low to high-temperature districts [27] (Appendix A). However, the mycelia can still maintain a high decomposition speed, which is similar to some insensitive varieties to temperature [28]. The research of the adaption mechanism of high temperatures was still absent, especially for a long-term response to high temperatures.

Here, we combined long-read sequencing technology, the Pacific Biosciences (PacBio, CA, USA) sequencing platform, and Hi-C (in vivo fixation of chromosomes) scaffolding techniques to obtain a high-quality chromosome-level genome assembly. The high-quality genome assembly provided an opportunity for genome structure research. A two-speed genome and accessory chromosome may develop resistance to harsh environments, which were revealed in macrofungi at the leading edge. The depiction of the oxalic acid synthetic pathway increased the understanding of wood degradation beyond simple CAZymes comparison. Transcriptome analyses under 25 °C and 33 °C provided a reference for high-temperature adaption and insensitive mechanism to a change of temperature. Further exploration of gene family and mating gene analyses provided more accurate evidence for the sexual and asexual process of *W. hoelen*.

## 2. Results

### 2.1. Genome Sequencing and Assembly

To achieve a high-quality genome assembly, the genomic DNA from the homokaryotic strain SS20 of *W. hoelen* [9] was extracted and sequenced. A de novo genome assembly was performed by the integration of PacBio single-molecule long-read sequencing, Illumina short-read sequencing, and Hi-C sequencing (Figure 1). A genome of 64.44 Mb was assembled from 1,497,235 PacBio reads (~330×, 21.02 G data size), which is smaller than the estimated genome size of 70.2–77.3 Mb from the k-mer analysis. A 78-contig draft genome was assembled consisting of a contig N50 of ~3.76 Mb, with the longest contig being 5.29 Mb (Appendix A). The genome size of *W. hoelen* is the biggest among the reported genome of the species of Polyporales (Appendix A). A total of 97.36% of 758 fungal BUSCO genes and 95.97% of 248 core genes by CEGMA were completely detected in the genome, indicating the completeness of the assembled genome (Appendix A). Furthermore, 22 assembled contigs (90.36% of the genome) were successfully clustered into 14 chromosome groups based on 47,710,330 raw reads (~110×, 7.07 G data) of Hi-C data. A heatmap created with Hi-C data suggested that all bins could be divided into the 14 chromosomes, which was consistent with the LACHESIS assembly (Figure 2A). Within each group, the intensity of the interaction at the diagonal position is higher than that of the off-diagonal position, indicating that the interaction between adjacent sequences (diagonal position) in the result of Hi-C assembly is high. However, the interaction signal strength between non-adjacent sequences (off-diagonal positions) is weak, which is consistent with the principle of Hi-C-assisted genome assembly. There is no obvious noise (stronger interaction intensity) outside the diagonal, which proves that the genome assembly effect is satisfactory. As a result, a total of 22 contigs (28.6%), 58.26 Mb (90.36%) sequences were anchored to the 14 chromosomes, with the longest chromosome being 7.47 Mb and the shortest 1.87 Mb (Figure 2A). All 14 chromosomes detected the telomeres at both ends (Figure 2B).

Specifically, a total of eight chromosomes were assembled through overlapping (Chr04, 06–12) and they were complete chromosomes. Three chromosomes (Chr02, 03, 14) have long repeat sequences in the connected positions with 100 bp “N” of contigs, which may be caused by continuous long sequence repeats. There were 5.5 Kb, 9.5 Kb, and 40 Kb long repeat fragment units for the three connected chromosomes, respectively. In addition, the Contigs of Chr01, Chr05, and Chr13 were directly connected with the 100 bp “N” without any characters (Figure 2B). All of the assembly of contigs was supported by the Hi-C interaction signals (Figure 2A). The breaks caused by long repeat fragments would be not easy to repair because of their large size and ambiguous copy numbers.

### 2.2. Gene Prediction and Functional Annotation

A large number of repetitive sequences were identified in *W. hoelen* strain SS20, accounting for 48.56% of the whole genome (Table 1). Among the different species, *W. hoelen* has a high ratio of repetitive sequences (Appendix A). About 29.8 Mb (46.26% of the genome) were identified as transposable elements, among which 36.76% were Class Ⅰ (retrotransposon) and 9.5% were Class Ⅱ (transposon). Retrotransposons consisted of 34.74% Class I long terminal repeats (LTR), 1.9% long interspersed nuclear elements (Line), and 0.12% short interspersed nuclear elements (Sine). The most abundant LTRs were *Gypsy* with 16.43 Mb (25.49% of the genome). Transposons included 5.5% DNA transposons, 3.72% miniature inverted-repeat transposable elements (MITE), and 0.23% rolling-circle (RC). Approximately 0.24% of the genome was identified as tandem repeats, with a total of 1584 SSRs identified (Table 1). The number of SSR was close to our previous prediction results of SSR number in strain 5.78 by MISA and Perl scripts (1612) [9].

A total of 10,465 genes were predicted based on de novo prediction (10,375), homology prediction (7260), and RNA-seq prediction (8644), with an average sequence length of 2004.04 bp. Each predicted gene contained 7.41 exons with an average length of 191.33 bp and 6.41 introns with an average length of 91.55 bp. A total of 9706 (92.75%) genes were annotated in five different databases (NR, GO, KEGG, KOG, and SwissProt) (Appendix A). A total of 10,319 (98.60% of all the predicted proteins) putative protein-coding genes were supported by RNA-seq data. At the same time, 31 secondary metabolite clusters were predicted, including the synthetic gene clusters of terpene, NRPS and T1PKS, which were also previously predicted by Cao et al. [24]. For the non-coding gene, 211 ncRNA were predicted, including 32 rRNAs, 28 snRNAs, 149 tRNAs, and 2 regulatory that occupied 0.0479% of the genome.

The genome of strain CGMCC 5.78, one of the most widely used strains in China, has been reported recently by Luo et al. [23], while Cao et al. [24] published a genome of a cultivated strain in Luotian county named WCLT. The comparison of these three assemblies indicated the best quality of the *W. hoelen* genome in this study was SS20 because of the fewest scaffolds and the longest N50 value of scaffolds (Table 2). The difference in the genome size, gene number, gene characters (intron and extron number and size), and transposon ratio may be affected by the sequencing technology and platform. However, the PacBio sequencing platform was used for the genomes sequenced by Cao et al. [24] and SS20 of this study, with there also being great differences. The strain used by Cao et al. [24] was the protoplast that has a lesser nuclei under a fluorescent microscope, with the homokaryon or heterokaryon not able to be determined. Here, the strain SS20 was confirmed to be homokaryon by culture characteristics, SSR marker, and heterozygosity analysis [9], which simplified genome assembly and provided a high-quality genome.

### 2.3. Whole-Genome Collinearity Analysis of Wolfiporia cocos, Wolfiporia hoelen, and Laetiporus sulphureus

To further evaluate the genome quality, the collinearity of genome SS20 and WCLT [24] was analyzed (Figure 3). In total, 78.58% genes have collinearity between the two genomes, without considering the gene prediction methods and real difference between two genomes, showing a high ratio of collinearity. That represents a high completeness of gene assembly, whereas Cao et al. [24] assembled some complete chromosomes into one new chromosome, and the telomeres were detected in the connected region with 100 bp “N” (Figure 3 and Appendix A). In total, 12 telomeres were discovered next to the 100 bp “N” inside different chromosomes, including 7 (CCCCTAA)n telomeres and 5 (GGGGTTA)n telomeres, among which 4 and 5 telomeres were detected in the first and second chromosomes which was consistent with the collinearity result (Figure 3). It was shown that there may be an incorrect chromosome number for the assembly of WCLT.

The genome of *W. cocos* was sequenced using a combination of ABI3703, 454-Titanium, and Illumina GAII, and showed a relatively complete genome [29]. In relative terms, 67.35% (7722) of proteins of *W. hoelen* have homologous proteins in *W. cocos*. There may have been a chromosome translocation from Chr7 to Chr13 of *W. hoelen* compared with *W. cocos* (Appendix A). The genome of *L. sulphureus* was sequenced by illumina that showed a relatively bad assembly [30] and a distant genetic relationship. In relative terms, 46.41% (5670) of proteins of *W. hoelen* have homologue in the genome of *L. sulphureus*. Only 29.34% genes on Chr14 of *W. hoelen* have collinearity with the species *W. cocos* and no homologous genes with *L. sulphureus* (0%); however, 70–80% of genes on the other chromosomes have homologue in *W. cocos* and 35–70% in *L. sulphureus*.

### 2.4. Is Chr14 Accessory Chromosome?

As shown in Figure 2B, the repetitive sequences were randomly distributed on different sites on chromosomes and the number of genes was negatively correlated to repetitive sequence sites (Figure 1). Chr14 is about 1.87 Mb and has the highest ratio of repetitive DNA sequences (71.56%) among all the chromosomes. A total of 167 genes were annotated on Chr14 and no single-copy genes were identified (Appendix A). Further analysis revealed that Chr14 has the lowest gene density (0.09/kb), lowest GC content (50%), fewest number of genes, lowest ratios of both homologous genes and single-copy homologous genes screened by OrthoFinder, and secreted proteins (Appendix A). It seems that Chr14 has the obvious characteristics of the accessory chromosome. A result of collinearity analysis, that there were no collinear proteins between Chr14 of *W. hoelen* and *L. sulphureus*, showed more evidence for the accessory chromosome Chr14 (Table 3).

KEGG analysis showed that the genes located on Chr14 were significantly enriched with phenylalanine, tyrosine, and tryptophan biosynthesis (ko00400), biosynthesis of amino acids (ko01230), amino sugar and nucleotide sugar metabolism (ko00520), and homologous recombination (ko03440) (Appendix A, *p* < 0.05). Phenylalanine, tyrosine, and tryptophan metabolism was involved in waterlogging and drought resisting of *Chenopodium quinoa* willd [31], while phenylalanine metabolism responded to cold stimulation in *Brassica napus* [32]. Phenylalanine and tyrosine metabolism was involved in disease resistance of *Gastrodia elata f. glauca* and rice [33,34]. Meanwhile, amino sugar and nucleotide sugar metabolism was also separately involved in drought resistance and disease resistance in *Reaumuria soongorica* and *Poa pratensis* [35,36]. Moreover, the secondary metabolism always happened in unsuitable environments, which suggests that Chr14 may have a relationship with the adaption of bad circumstances. The high ratio of transposon, low gene diversity, low GC content, absence of single-copy homologous genes, and resistance genes gathered on Chr14 showed that Chr14 was the accessory chromosome and the genome of *W. hoelen* was a two-speed genome.

### 2.5. Whole-Genome Methylation and Methylation Level of Transposons

Whole-genome 5-methylcytosine (m5C) level was determined. A 5.78% methylation level of cytosines was revealed in *W. hoelen* strain SS20. According to the sequence background of cytosine, DNA cytosines can be divided into three types: CG, CHG, and CHH (H  =  A, C, or T). Different contexts of methylation sites accounted for different proportions in the whole genome, of which mCG sites accounted for the largest proportion (93.48%), followed by mCHH (5.18%) and mCHG (1.34%) sites (Appendix A). This result indicates that mCG contexts of methylation sites were dominant in this species. The adjoining nucleotide sequences were considered to have a relationship with the methylation level, but there were no conserved sequences detected around mCG, mCHG, and mCHH.

The methylation sites are highly consistent with the distribution of transposons on the chromosomes (Figure 4A). The methylation levels of the TE body, the 2 kb region upstream and downstream of the TE body, were counted. A high ratio of methylation appeared in the upstream and downstream regions of the TE body, including mCG, mCHG, and mCHH (Appendix A). That suggests the function of transposon was restrained, which largely ensures the stabilization of the genome.

The expression levels between methylated genes and unmethylated genes were compared. The average expression level of unmethylated genes is higher than that of methylated genes (Figure 4B). The methylation ratio and the expression level showed a weak negative correlation (−0.34) (Figure 4C). The methylation ratio on transposons was 15.09%, much higher than that on genes (1.28%); however, there was no obvious GC bias, with 52% GC content for transposons and 55% for genes. The high ratio of methylation has a strong regulation ability on the function of transposons, which play a role on gene transfer among the genome. Thus, the methylation level of transposons directly affects the gene transfer, which would function on auto-regulation for genome stability and adaptive evolution.

### 2.6. Phylogenetic Analysis and Gene Family Expansion Associated with Sclerotia

A maximum likelihood (ML) phylogeny analysis for *W. hoelen* and the 11 additional fungal species was performed based on the shared single-copy orthologous genes, which were concatenated into a supermatrix with 659,265 amino acid sites. The clade of Polyporaceae was separated from the others supported with bootstrap values of 100%. *W. hoelen* was phylogenetically close to *W. cocos* and then grouped with *Laetiporus sulphureus* (Figure 5A). They are Laetiporaceae fungi, which confirmed the revision of the family-level classification of the Polyporales by Justo et al. [37] based on three genes: *rpb1*, nrLSU, and nrITS. Phylogenomic analysis supported the different species of *W. cocos* and *W. hoelen.*

The expanded genes were manually counted according to the orthogroups predicted with the OrthoFinder. Compared with other species in Polyporales, most of the gene families of *W. hoelen* have relatively low copies (Appendix A). There are many gene families with more than 100 copies in *Ganoderma sinensis*, *L. sulphureus*, *Obba rivulosa*, etc. (Appendix A); however, the maximum number of copies is 23 in *W. hoelen*, with only 24 families having more than 10 copies. The top expanded gene families in the *W. hoelen* genome were identified. These families were primarily associated with signal transduction, carbohydrate transport, and metabolism; secondary metabolites biosynthesis, transport, and catabolism; lipid transport and metabolism; intracellular trafficking, secretion, and vesicular transport; and cell cycle control, cell division, and chromosome partitioning functions, with almost half of them having no specific functional annotation (21/38) (Appendix A). Because *W. hoelen* and *W. cocos* can form sclerotia, the common specific expanded gene families were also analyzed; it was found that the function of common expanded families was similar in these two species (Figure 5B).

### 2.7. Mating Locus of W. hoelen

James et al. [38] determined the bipolar mating system of *W. cocos*, using the criterion that DNA polymorphism of *MAT* genes should be extreme. The bipolar mating system of *W. hoelen* was recently confirmed in this laboratory according to the mating tests [26]. The mating was controlled by the *MAT-A* locus (*HD*: homeodomain) in bipolar basidiomycetes, with the *MAT-A* locus analyzed based on the genome sequences of *W. hoelen*.

One pair of *HD1* and *HD2* genes and a single *HD3* of *W. hoelen* were annotated, with all located on Chr02. In addition, the location of the *HD1-HD2* pair and *HD3* in the same chromosome was first demonstrated. The corresponding *HD1*, *HD2*, and *HD3* were detected through blastp in *W. cocos* (MD-104 SS10, SRR061042). Being different from *W. cocos* and *W. hoelen*, *L. sulphureus* has three *HD1* genes and a single *HD2* gene, with no other genes between the *β-fg* and *MIP*. HD1 and HD2 have low protein identity (HD1 36.98% vs. *W. cocos*, HD2 38.15% vs. *W. cocos*), whereas surrounding proteins and HD3 have high identities (vs. *W. cocos*, ≥95%) (Appendix A). That implies the genes *HD1* and *HD2* probably play the roles on mating recognition, but *HD3* does not function on it.

Further, the three-dimensional structure and conserved domain of HD proteins were analyzed. The conserved domain and motif (WFXNXR) were detected in HD2 and HD3 (Figure 6B,C), but were not detected in HD1, while the three-helix DNA binding motif was also not detected (Figure 6A). That showed the big variation of HD1 in different species, which is less conserved than HD2. Phylogenetic analysis of proteins HD1 and HD2 among different species showed the great genetic distance of HD in *W. hoelen* with other species (Appendix A).

There was a great difference in the genes located between *β-fg* and *HD1* in *W. hoelen* and *W. cocos*. An approximately 30 kb long repetitive fragment (33,424 bp), which was absent in *W. cocos* and *L. sulphureus*, was discovered between *HD1* and the upstream gene in *W. hoelen.* The long repeat sequences were mainly transposons (75.6%) (Appendix A), including 59.1% LTR, 7.3% MITE, 9.2% DNA, and 5.7% unknown repeat sequences. The accumulation of transposon and the existence of non-homologous genes would cause recombination suppression in all possibilities [39], which may further affect the expression of *HD1* and *HD2* [40]. *rpb2* was located on the 82,184 bp upstream of *HD1*, which provided genome-level evidence of the possible linkage between *rpb2* and *MAT-A* loci discovered in our previous research [26].

The *MAT-B* locus was also analyzed. Four, four, and two pheromone receptors were annotated in the genome of *W. cocos*, *W. hoelen*, and *L. sulphureus*, respectively, which all have seven transmembrane domains as expected (Appendix A). Three pheromone precursors were detected around the pheromone receptor through ORF prediction and conserved motif analysis based on gene annotation. No pheromone precursor was detected in the *L. sulphureus* genome and six were annotated in the genome of *W. cocos*. The pheromone receptor proteins were more conserved than HD proteins, with a high identity of 94–99% compared with *W. cocos* (Appendix A). That was in accordance with the bipolar mating system.

### 2.8. CAZymes Family Comparison and Oxalic Acid Synthetic Pathway

A total of 214 CAZymes were identified in *W. hoelen*, which was similar to the species *W. cocos* (205) in the same genus, and only small differences were discovered in some families. A comparative analysis of CAZymes of *W. hoelen* was conducted with eighteen other wood-rotting fungal species in Polyporales, including four WB (white-rot, on the broadleaf tree) species, five WB/C (white-rot, on the broadleaf tree or coniferous tree) species, five BC (brown-rot, on the coniferous tree) species, and four WC (white-rot, on the coniferous tree) species (Appendix A) [41,42,43]. It was found that the CAZymes were less than almost all the white-rot species, no matter the growth on broadleaf or coniferous trees (Appendix A).

Mann–Whitney U test results showed white-rot (WB, WC, and WB/C) species were equipped with more CAZymes than brown-rot (BC) species in all six families, and those species which can grow on broadleaf trees (WB and WB/C) possess more CAZymes than those that only grow on coniferous species (WC and BC) in all families (*p* < 0.05) (Appendix A). In addition, the specific differential CAZymes genes were depicted in Appendix A. GH55 was the only gene that showed more than white-rot species in brown-rot species which have exo and endo-β-1.3-glucanase activities. The amount of CAZymes in the AAs and GHs family has a significant difference between white-rot and brown-rot species (Appendix A), while the difference was also discovered between species grown on broadleaf trees and those only grown on coniferous trees (Appendix A). Comparing the average number of different families in the WB, WB/C, WC, and BC species, the trend was as follows: WB and WB/C > WC > BC. In agreement with earlier findings, the genomes of brown-rot fungi are mostly devoid of known ligninolytic genes (Appendix A), which have important functions in lignin decomposition. The characteristics of CAZymes of *W. hoelen* were apparently consistent with that of brown-rot fungi.

Both *W. hoelen* and *S. crispa* were brown-rot fungi and cultivated on a large scale in China with pine sawdust substrates. However, the growth rate of *W. hoelen* on pine sawdust substrates was almost 2–3 times that of *S. crispa* (Appendix A). The CAZymes of these two species were compared. The results showed that the number of CAZymes for different decomposition constitutes [44,45] in *W. hoelen* was more than that in *S. crispa*, which may explain the extremely different growth rates on pine sawdust substrates (Appendix A, Appendix A).

In addition, *W. hoelen* was always studied as a copper-tolerance fungus that can produce massive oxalic acid, which has important functions on wood decomposition, especially for cellulose and pectin [46,47]. Except for complexes with iron Fe^3+^ to promote reduction, oxalic acid can play a role in wood degradation [46]. Oxalic acid is likely to act synergistically with polygalacturonase to solubilize and hydrolyze the pectin in pit membranes and middle lamellae. The production of oxalic acid and polygalacturonase should facilitate the early spread of hyphae and enhance the lateral flow of wood-decay enzymes and agents into adjacent tracheids and the wood cell wall [47]. Here, we annotated all the genes involved in oxalic acid and depicted the synthetic pathway of oxalic acid in *W. hoelen* (Figure 7A). In addition, the acid production characteristics at different stages were also detected. Mycelia of *W. hoelen* are suitable for growth under pH 3–5 in liquid fermentation (Figure 7B), and the mycelia can decrease the pH to 3–4 from 7 after being cultured for only 1 day (Figure 7C). When the sclerotia formed and fruited, the pH was also kept at 3–4 (Figure 7D,E). That showed the acid production may not only function on wood-rotting, but also play important roles in the growth of mycelia, sexual fruiting body formation, and asexual sclerotia formation. At the same time, the pH may affect the TCA cycle. On the one hand, the pH can affect the activity of enzymes involved in the TCA cycle; on the other hand, the oxalic acid synthesis would be inhibited, with oxaloacetic acid more used to TCA which would contribute more energy for other processes.

### 2.9. High-Temperature Adaption of W. hoelen Based on Transcriptome Analysis

To explore the high-temperature adaption of *W. hoelen*, comparative transcriptome analysis was performed between the mycelia of strain CGMCC 5.545 cultured under 25 ℃ (normal temperature, NT) and 33 °C (highest tolerance temperature, HT) continuously. There were 830 and 1060 genes (more than 2 of 3 replicates, TPM < 1) not expressed under 25 °C and 33 °C, respectively. Of 10,567 genes, 3471 (32.85%) showed different expressions, among which 549 were down-regulated and 2922 were up-regulated when NT was compared with HT (Appendix A). Pathway analysis showed that the DEGs were enriched in various metabolism and processes, mainly in global and overview maps of metabolism that showed temperature has a wide regulation of life activities (Appendix A). Combining the widely down-regulated glycolysis and TCA cycle (Appendix A), the growth of mycelia under 33 °C should be badly suppressed.

However, there was no significant difference for the growth rates and mycelial biomass cultured under 25 °C and 33 °C (Figure 8A). The pathway analysis showed that up-regulated genes were mainly enriched in glutathione and lipid metabolism (Appendix A). The genes encoding xyloglucan-specific endo-beta-1,4-glucanase, endoglucanase, xylanase, and mannan endo-1,4-beta-mannosidase A were upregulated 10–100 fold, implying the change of degradation components which may provide the materials for hyphae growth and cell wall synthesis (Figure 8C) under continuous high temperatures. Glucan and chitin were the chief components of the cell wall of basidiomycetes, but all the glucan synthase and chitin synthase were significantly down-regulated, implying the components of the cell wall may experience some changes under high temperature (Appendix A). The up-regulation of genes encoding glycerolipid dehydrogenase and biosynthesis of unsaturated fatty acids would be essential for keeping fluidity of the cytomembrane and normal cell activities under high temperatures (Appendix A). The main ROS clearance genes, GSH (glutathione), SOD (superoxide dismutase), and CAT (catalase), showed significant up-regulation, indicating that more reactive oxygen species would be produced under 33 °C. The ROS staining also supports the result of ROS clearance genes (Appendix A). Overall, the up-regulation expression of genes for distinctive decomposition enzymes, ROS clearance, biosynthesis of unsaturated fatty acids, and change of the cell wall components which can maintain high-speed growth of mycelia, may be the high-temperature adaption strategy of *W. hoelen*.

In previous studies, the short high-temperature stimulation improved the content of ganoderic acid in *Ganoderma lucidum* [48]. Therefore, the triterpene synthetic genes were analyzed here, but only IDI (isopentenyl diphosphate isomerase) was up-regulated, with the majority of genes involved in the triterpene synthetic pathway down-regulated, including HMGS (3-hydroxy-3-methylglutaryl-CoA synthase), HMGR (3-hydroxy-3-methylglutaryl-CoA synthase), LS (lanosterol synthase), MVK (mevalonate kinase), PMK (phosphomevalonate kinase), MVD (pyrophosphomevalonate decarboxylase), and SE (squalene monooxygenase) (Appendix A). It was inferred that continuous high temperatures may have different effects than short high-temperature simulation. Additionally, the expression levels of *hsp* genes were analyzed. The majority of *hsps* were down-regulated, with only the small heat shock protein C4 showing up-regulation (Appendix A), which differs from the response of short high-temperature simulation [48].

## 3. Discussion

Sclerotia of *W. hoelen* generated irreplaceable effects in many prescriptions and are extensively used in China as well as other East Asian countries. Here, we assembled the first high-quality homokaryotic genome of *W. hoelen* combining PacBio sequencing and Hi-C scaffolding technology. The genome structure, transposon and methylation, high-temperature adaption, gene family expansion about sclerotia, and mating genes were analyzed here. The results could provide a reference for further studies of specific gene function and regulation, genetics, epigenetics, genome structure variation, and evolution.

### 3.1. Ploidy, Chromosome Number, Telomere, Heterozygosity, and Genome Assembly

The hyphae of *W. hoelen* were deficient of clamp connection [49], resulting in great difficulties in distinguishing the homokaryon, which is important for accurate chromosome-level genome assembly. Differing from the two previous genomes, which were sequenced with heterokaryotic strain and illumina platform [23], and with a wrong chromosome number [24], the strain sequenced here was definitely a homokaryotic strain which was previously demonstrated and conforms to our previous mating research [9,26]. Ploidy was the most important index to distinguish chromosome number. Actually, for basidiomycetes, the mycelial stage is haploid and the diploid stage is normally confined to the basidia, in which the two different nuclei fuse and form the diploid nucleus (Figure 9). The nuclear number can be analyzed using the protoplast by flow cytometry analysis and we cannot obtain a diploid conclusion of *W. hoelen*, but the protoplasts or nuclei can be used for ploidy analysis in plants. Combining the observation results of chromosomes under a fluorescence microscope [24], the haploid genome of *W. hoelen* should consist of 14 chromosomes with each having different lengths (Figure 2A). On the other hand, Cao et al. [24] reported the 14 chromosomes as 7 pairs based on the similar size of every 2 chromosomes; however, the picture is not clear enough to divide into 7 pairs [24].

Each chromosome has 2 telomeres in the terminal, with 28 telomeres detected in this study, which was identical to the 14 chromosomes observed under a fluorescence microscope [24]. The telomeres of 7 chromosomes assembled by Cao et al. [24] were analyzed and it was found that the telomeres in the terminal were absent, with many telomeres appearing in the interior of chromosomes and adjoining with “N”. The bad genome quality and wrong chromosome number was also shown.

Heterozygosity represents the purity of the genome, which is always evaluated with the original reads. Our previous research has reported that the heterozygous ratio of the genome of heterokaryotic strains of *W. hoelen* was approximately 0.75% and that of the homokaryotic strain was only 0.01% [9]. The heterozygosity of Cao et al. was absent [24], with a high heterozygous ratio affecting the genome assembly.

### 3.2. Two-Speed Genome and Homeostasis

The two-speed genome was discovered in pathogenic fungi which always had a relationship with adaptive evolution [25]. The two-speed genomes have been revealed in *Fusarium graminearum*, *Verticillium dahliae*, *Phytophthora infestans* (Oomycetes), etc. [50,51,52]. In the filamentous pathogen genomes, the gene sparse, repeat rich, and virulence-associated genes clustered compartments served as a cradle of adaptive evolution which led to a two-speed genome model. The probable two-speed genome of *W. hoelen* was revealed, and the accessory genome Chr14 may function widely in resistance to bad environments and diseases. There was no two-speed genome of edible or medicinal mushrooms reported to our knowledge, which was mainly attributed to the simple genomes and limitation of the sequencing technology. Some symbiotic macrofungi also showed significant transposon expansion. Wu et al. [53] discovered that the symbiotic species showed transposon expansion more than saprotrophic species in Boletales. The revealing of two-speed in *W. hoelen* was at the leading edge in macrofungi and would promote the research of complex genomes in macrofungi.

Transposons are the main factor affecting genome stability. For restraining the function of transposons and maintaining the stability of the genome, transposons may have universal methylation [54,55]. Here, different from low methylation on cytosine in the majority of fungi [56], a 5.78% methylation level of cytosines was revealed in *W. hoelen*, while absolutely no m5C methylation in *Saccharomyces* spp. was detected [57,58,59]. In vertebrates and plants, transposons always have high-level methylation [58], which represented the senior regulation of transposons which may be expanded through great environmental change. That also implies *W. hoelen* has precise regulation of genome stability, but the variability of homokaryotic offspring may show the opposite aspects of the effects of transposons on gene expression regulation. It was reported that *Morchella sextelata* had 0.42% m6A methylation level and also showed a preference for transposons [60]. The study of epigenetics in macrofungi is still at the primary stage, which would be pay more attention for revealing biological process.

### 3.3. High-Temperature Adaption Strategy

*W. hoelen* is mainly cultivated in regions with a subtropical and humid climate, such as China, Vietnam, and Thailand. In China, the sclerotia of *W*. *hoelen* have been cultivated for more than 1500 years and the cultivation has a relatively clear migration history of cultivation districts [26] (Appendix A). That may provide clues for high-temperature adaptive evolution in different cultivation districts of *W. hoelen*.

Many varieties of cultivated macrofungi were bred for adaption to different cultivation environments, with the strains often divided into sensitive and insensitive to temperature categories [27]. For the high-temperature tolerance mechanisms, Liu et al. [27] explored the protective roles of trehalose in *Pleurotus pulmonarius* during the heat stress response, including the change of content, exogenous application, and related gene expression. Zhang et al. [61] studied the response of heat shock protein, mycelial growth, and hyphal branching in *Ganoderma lucidum* to heat stress. However, all the related studies were conducted in relation to the response to a short high-temperature stimulation, with an absence of the study of long-term response. 

In this study, the protection reaction from up-regulation of ROS cleavage genes and biosynthesis of unsaturated fatty acids, which also respond to short heat stimulation and change of environments [62,63]. The change of the cell wall components was also supported by down-regulation of chitin synthase and glucan synthase, which was consistent with the results that carbon source, pH, temperature, aeration, and culture modes affect the cell wall composition of *Saccharomyces cerevisiae* [64]. Unexpectedly, however, the high-fold up-regulation of distinctive decomposition enzymes caught our attention, implying that regulation of different degradation enzymes under different temperatures was the high-temperature adaption strategy with the help of ROS cleavage, change of cell wall, and various protection reactions. This may also provide a reference for the study of the insensitive mechanism of specific varieties.

Differing from the short heat stimulation, the majority of the *hsp* genes, triterpene synthetic genes, and trehalose-related genes were down-regulated with the decline of the global metabolic level under 33 °C for long-term culture compared with 25 °C. It was indicated by the different adaption mechanisms between long-term and short stimulation of high temperatures, but the level of reactive oxygen species improved under both conditions [65] and the genes encoding for ROS cleavage enzymes, SOD, CAT, and GSH, were significantly up-regulated. In addition, part of the mycelia culture under 33 °C changed to brown quickly, which may be induced by high ROS levels. The protein is over-presented in mycelial brown film response to ROS in *Lentinula edodes* [66], which may support our conjecture.

### 3.4. Gene Family Expansion for Sclerotial Development

Different to the majority of edible and medicinal mushrooms, the sclerotia were the edible and medicinal part of *W. hoelen* and was paid more attention. Zhang et al. [20] discovered that 69 CAZyme-encoding genes were significantly up-regulated at the early stage of sclerotial growth, with more than half belonging to the GH family, indicating the importance of the GH family for degrading the pine woods. Luo et al. [23] presented 21 up-regulated and 13 down-regulated signaling genes that may play a key role in the developmental transition from mycelium to sclerotium combining sclerotial development genes studied in *S. sclerotiorum* and differentially expressed genes between mycelia and sclerotia. According to Cao et al. [24], the wood decomposition enzymes, as well as sclerotium-regulator kinases, aquaporins, and highly expanded gene families such as NAD-related families, together with actively expressed 1,3-β-glucan synthase for sclerotium polysaccharides, contributed to the sclerotium formation and expansion. 

Here, we discovered that the significantly expanded gene family was enriched in signal transduction, nutritional transport, and vesicular transport. Signal transduction and a mitogen-activated protein kinase pathway were deemed as important formation mechanisms of macrofungal sclerotia [67]. Wei et al. [21] discovered several protein kinase-regulated MAPK signal pathways that participated in the conversion from mycelia to sclerotia. The transport, biosynthesis, and metabolism of secondary metabolites, carbohydrates, and lipids may also be an important part of sclerotial formation, which was the structure for nutrition storage. Given sclerotial formation is a complex process that is regulated by many genes, the specific key genes should be further verified through gene knockout or editing.

## 4. Materials and Methods

### 4.1. Homokaryotic Strain Screening and DNA Preparation

The strain CGMCC 5.545 was collected from the CGMCC (China General Microbiological Culture Collection Center) and had been preserved in 1991, with the homokaryotic strain CGMCC 5.545 SS20 isolated and identified by our previous study [9], and used for genome sequencing (Figure 10). The strain SS20 was cultured in liquid media (glucose 20.0 g, yeast extract 4.0 g, peptone 5.0 g, KH_2_PO_4_ 1.0 g, MgSO_4_ 0.5 g, VB_1_ 1.0 mg/1000 mL, pH 3.0) at 25 °C and 150 rpm for 10 d. The mycelia were collected by centrifugation at 4000 rpm/min for 10 min, washed twice with sterile water, and stored at −80 °C until DNA extraction. Total DNA extraction and quality evaluation followed the methods of Li et al. [9].

### 4.2. Genome Sequencing and Assembly

The Illumina NovaSeq platform and PacBio Sequel platforms were used for genome sequencing at Nextomics Biosciences Co., Ltd. (Nextomics Biosciences, Wuhan, China). The Illumina sequencing results have been reported in our previous work [9]. A 20 Kb insert DNA library was constructed according to PacBio’s standard protocol (Pacific Biosciences, CA, USA). Briefly, a total amount of 2 µg DNA was used for the DNA library preparations. The genomic DNA sample was cut by g-TUBEs (Covaris, MA, USA) according to the expected fragment size for the library. Single-strand overhangs were then removed and DNA fragments were damage-repaired, end-polished, and ligated with the stem-loop adaptor for PacBio sequencing. Link-failed fragments were further removed by an exonuclease and target fragments were screened by the BluePippin (Sage Science, MA, USA). The SMRTbell library was then purified using AMPure PB. Agilent 2100 Bioanalyzer (Agilent technologies, CA, USA) was used to detect the size of library fragments. Sequencing was performed on a PacBio Sequel II instrument with Sequel II Sequencing Kit 2.0 on Nextomics Biosciences (Wuhan, China).

Raw sequencing data produced by Pacific Bioscience Sequel were processed following the quality control protocol through the SMRT Link v8.0 to remove low-quality reads and adapters resulting in subreads. After obtaining the subreads, the genome was assembled de novo into contigs by using an OLC (overlap-layout-consensus) with Falcon. The completeness of genome assembly was assessed using BUSCO v4.0.5 (Benchmarking Universal Single Copy Orthologs, EZlab, Geneva, Switzerland) and CEGMA v2.5 (Core Eukaryotic Gene Mapping Approach. Korf Lab, CA, USA).

### 4.3. Hi-C Sequencing and Assembly

Telomere sequences of (CCCCTAA)n or (TTAGGGG)n [68] were analyzed and the chromosome number was predicted according, primarily, to the telomere number. To anchor hybrid scaffolds onto the chromosome, genomic DNA was extracted for the Hi-C library from *W. hoelen* strain SS20. The Hi-C library was then constructed and sequencing data were obtained from the Illumina Novaseq platform (Illumina, Inc., CA, USA). In brief, freshly harvested mycelia were vacuum infiltrated in nuclei isolation buffer supplemented with 2% formaldehyde. Crosslinking was stopped by adding glycine and additional vacuum infiltration. Fixed tissue was then grounded to powder before resuspending in a nuclei isolation buffer to obtain a suspension of nuclei. The purified nuclei were digested with 100 units of DpnII and marked by incubating with biotin-14-dCTP. Biotin-14-dCTP from non-ligated DNA ends was removed owing to the exonuclease activity of T4 DNA polymerase. The ligated DNA was sheared into 300–600 bp fragments and was then blunt-end repaired and A-tailed, followed by purification through biotin–streptavidin-mediated pulldown. Finally, the Hi-C libraries were quantified and sequenced using the Illumina Novaseq platform at Nextomics Biosciences Co., Ltd. (Wuhan, China).

In total, 47.7 million paired-end reads were generated from the libraries. Quality control of Hi-C raw data was then performed using Hi-C-Pro as former research. Firstly, quality scores <20, and sequences shorter than 30 bp were filtered out using fastp, then the clean paired-end reads were mapped to the draft-assembled sequence using bowtie2 (v2.3.2) (-end-to-end --very-sensitive -L 30) to obtain the unique mapped paired-end reads. Valid interaction paired reads were identified and retained by HiC-Pro v2.8.1 [69] from unique mapped paired-end reads for further analysis. Invalid read pairs, including dangling-end, self-cycle, re-ligation, and dumped products, were filtered by HiC-Pro v2.8.1. The scaffolds were further clustered, ordered, and oriented onto chromosomes by LACHESIS [70]), with parameters CLUSTER_MIN_RE_SITES = 100, CLUSTER_MAX_LINK_ DENSITY = 2.5, CLUSTER NONINFORMATIVE RATIO = 1.4, ORDER MIN N RES IN TRUNK = 60, ORDER MIN N RES IN SHREDS = 60. Finally, placement and orientation errors exhibiting obvious discrete chromatin interaction were manually adjusted.

### 4.4. Repeat Annotation, Gene Prediction, Gene Function, and Noncoding RNA Annotation

Tandem repeats were annotated using the software GMATA v2.1 [71] (Genome-wide Microsatellite Analyzing Toward Application) and Tandem Repeats Finder (TRF, Benson, MA, USA). GMATA identifies the simple repeat sequences (SSRs) and TRF recognizes all tandem repeat elements in the whole genome. Transposable elements (TE) in the *W. hoelen* genome were then identified using a combination of ab initio and homology-based methods. Briefly, ab initio repeat library for *W. hoelen* was first predicted using MITE-hunter (Han and Wessler, GA, USA) and RepeatModeler v1.0.11 (Hubley, WA, USA) with default parameters, in which LTR_FINDER, LTRharverst, and LTR_retriver were also included for the genome. The obtained library was then aligned to TEclass Repbase [72] to classify the type of each repeat family. For further identification of the repeats throughout the genome, RepeatMasker was applied to search for known and novel TEs by mapping sequences against the de novo repeat library and Repbase TE library. Overlapping transposable elements belonging to the same repeat class were collated and combined.

Three independent approaches, including ab initio prediction, homology search, and reference-guided transcriptome assembly, were used for gene prediction in a repeat-masked genome. In detail, GeMoMa v1.3.1 [73] was used to align the homologous peptides from related species to the assembly and then obtain the gene structure information, which was homolog prediction. For RNAseq-based gene prediction, filtered mRNA-seq reads were aligned to the reference genome using STAR (default parameter). The transcripts were then assembled by StringTie and open reading frames (ORFs) were predicted using PASA. Augustus with default parameters were utilized for ab initio gene prediction. Finally, EVidenceModeler (EVM) was used to produce an integrated gene set of which genes with TE were removed using the TransposonPSI package [74] (http://transposonpsi.sourceforge.net/ (accessed on 15 November 2020)) and the miscoded genes were further filtered. Gene function information and motifs and domains of their proteins were predicted by comparing with public databases including SwissProt, NR, KEGG, KOG, and Gene Ontology (GO). The putative domains and GO terms of genes were identified using the InterProScan 5 with default parameters. For the other four databases, BLASTp was used to compare the Evidence Modeler-integrated protein sequences against the four well-known public protein databases with an E value cutoff of 1 × e^−05^, and the results with the hit with the lowest E value were retained. The results of the five databases were finally integrated.

Two strategies were used to predict ncRNA (non-coding RNA): searching against a database and predicting with the model. Transfer RNAs (tRNAs) were predicted using tRNAscan-SE [75] with eukaryote parameters. MicroRNA, rRNA, small nuclear RNA, and small nucleolar RNA were detected using Infernal version 1.1.2 to search the Rfam database 14.8 [76]. The rRNAs and their subunits were predicted using RNAmmer-1.2 [77].

### 4.5. Transcriptome Sequencing and Analysis

The mycelia cultured on PDA media covered with cellophane at 25 °C and 33 °C were used for RNA sequencing. RNA quality was evaluated with agarose gels, a spectrophotometer, and Agilent 2100, with RNA concentration measured using Qubit 2.0 (Thermo Fisher Scientific, MA, USA). A total amount of 2 µg RNA per sample was used as input material for the RNA sample preparations. Sequencing libraries were generated using VAHTS mRNA-seq v2 Library Prep Kit (Vazyme Biotech Co., Ltd., Nanjing, China) for Illumina following the manufacturer’s recommendations and index codes were added to attribute sequences to each sample. The libraries were sequenced on an Illumina NovaSeq platform to generate 150 bp paired-end reads, according to the manufacturer’s instructions.

Raw data of fastq format were firstly processed through primary quality control. In this step, clean data were obtained by removing read pairs that contain N more than 3 or the proportion of bases with a quality value below 5 is more than 20% in any end, or adapter sequence was founded. All downstream analyses were based on clean data with high quality.

Paired-end clean reads were aligned to the reference genome using Hisat 2, with featureCounts used to count reads number aligned to different genes. The TPM of different genes was calculated in WPS-Excel, with DEG (differential expression genes) analyzed with Omicshare Tools (GENE DENOVO) and a cluster heatmap built on BMKcloud.

### 4.6. Methylation Sequencing

Genomic DNA was isolated from the cultured mycelia with a Plant Tissue DNA Kit (OMEGA Bio-tek Inc., Norcross, GA, USA) according to the manufacturer’s instructions, with quality control subsequently performed on the purified DNA samples. Genomic DNA was quantified by using a TBS-380 fluorometer (Turner BioSystems Inc., Sunnyvale, CA, USA). A high-qualified DNA sample (OD260/280 = 1.8~2.0, >6 ug) was utilized to construct a fragment library.

Before bisulfite treatment, 25 ng lambda DNA were added to the 5 µg genomic DNA of *W. hoelen*. The mixed DNA was then fragmented with a Sonicator (Sonics & Materials, Newtown, CT, USA) to 300 bp. After blunt ending, 3′-end addition of dA, Illumina methylated adapters were added according to the Illumina manufacturer’s instructions. The bisulfite conversion of *W. hoelen* DNA was performed using the ZYMO EZ DNA Methylation-Gold kit (ZYMO) and amplified by 12 cycles of PCR. Ultra-high-throughput pair-end sequencing was performed using the Illumina Hiseq2500 (Illumina, Inc., CA, USA) according to the manufacturer’s instructions. Raw Hiseq sequencing data were processed by Illumina base-calling pipeline (SolexaPipeline-1.0). After the DNA sample was delivered, the quality was tested first. The qualified DNA sample was then used to construct the BS (Bisulfite sequencing) library. Finally, the qualified BS library was used for sequencing. The raw paired-end reads were trimmed and quality controlled by Trimmomatic with default parameters [78]. All clean BSseq reads were mapped to the reference genome with the BSMAP aligner allowing up to two mismatches [79]. Uniquely mapped reads were used to determine the cytosine methylation levels as previously stated [80].

### 4.7. Prediction of Secreted Proteins

Secretory proteins were identified based on recognizing the signal peptide and no transmembrane sequences. Proteins were deemed as secreted proteins if the signal peptides were identified by SignalP 4.0 [81] and TargetP 1.1 [82], while transmembrane sequences were not identified by at least one of the methods between TMHMM 2.0 [83] and WoLF PSORT [84].

### 4.8. Phylogenomic Analyses

Phylogenomic analyses were performed using 11 genomes of fungi belonging to five families of Polyporales (Appendix A). All the single-copy orthologous genes of 11 species were extracted using OrthoFinder (Steve Kelly Lab, Oxford, UK) [85]. The amino acid sequences of the single-copy genes were aligned using mafft version 7.505 with default parameters [86], with conserved sequences screened using Gblock [87] and concatenated to supergenes in Phylosuite [88]. Maximum likelihood-based phylogenetic trees were built using RAxML-NG version 0.9.0 [89], with the amino acid replacement matrix LG + F+I + G4 selected by Phylosuite. The agaricales *Laccaria bicolor* was used as the outgroup. One thousand bootstrap replicates were used and trees were figured in FigTree v1.4.3 (Rambaut, Edinburgh, UK) [90].

### 4.9. Whole-Genome Collinearity Analysis

To explore the genomic structure variation, whole-genome collinearity was frequently used. The collinearity between SS20 and WCLT, and *W. cocos*, *Laetiporus sulphureu*s which are in the same family Laetiporaceae, were analyzed and the whole 14 pseudo- chromosomes of *W. hoelen* were aligned to *W. cocos* (NCBI: PRJNA52943) and *L. sulphureus* (NCBI: PRJNA196057) using TBtools [91].

### 4.10. CAZymes (Carbohydrate-Active Enzymes) Family Annotation

The CAZymes in the *W. hoelen* genome were identified using the dbCAN2 [92] with three tools: E-value < 1 × e^−15^ and coverage > 0.35 for HMMER + dbCAN; E-value < 1 × e^−102^ for DIAMOND + CAZy; the number of conserved peptide hits > 6 and the sum of conserved peptide frequencies >2.6 for Hotpep + PPR. Those with # of Tools ≥ 2 were kept as candidates. The difference analysis of CAZymes number among basidiomycetes with different habitat types was determined by the Mann–Whitney U test (every two groups) at *p* = 0.05 by an independent sample of the non-parametric test using SPSS statistics 23 (International Business Machines Corporation, Armonk, NY, USA). Cluster heatmap figures were generated using the R project on the BMKCloud platform. The CAZymes of different decomposition constitutes were classified according to previous reports [44,45].

### 4.11. Conserved Protein Domain and Motif Prediction

The three-dimensional structure of the conserved domain was predicted in SWISS-MODEL [93], with plotting of the conserved motif performed on Weblog [94]. Multiple sequence alignments of the protein sequences and corresponding DNA sequences were performed using DNAMAN (Version 6) software (LynnonBiosoft, San Ramon, CA, USA). In addition, multiple sequence alignment results were displayed by MS Paint, software for Windows version 10 (Microsoft, WA, USA).

## Figures and Tables

**Figure 1 ijms-23-10484-f001:**
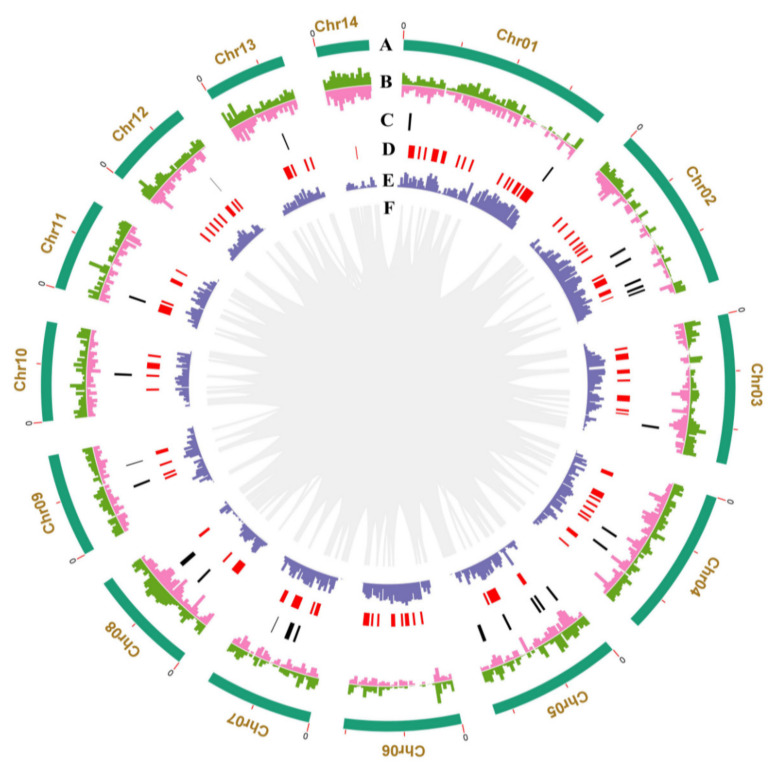
Circos graph of genome characteristics of *Wolfiporia hoelen*. (**A**) 14 pseudo-chromosomes; (**B**) Transposable elements (TEs) of which transposons are in green and retrotransposons in pink; (**C**) Clusters of secondary metabolites; (**D**) Non-coding RNA (ncRNA); (**E**) Gene density; (**F**) Large fragment duplication (>10 Kb).

**Figure 2 ijms-23-10484-f002:**
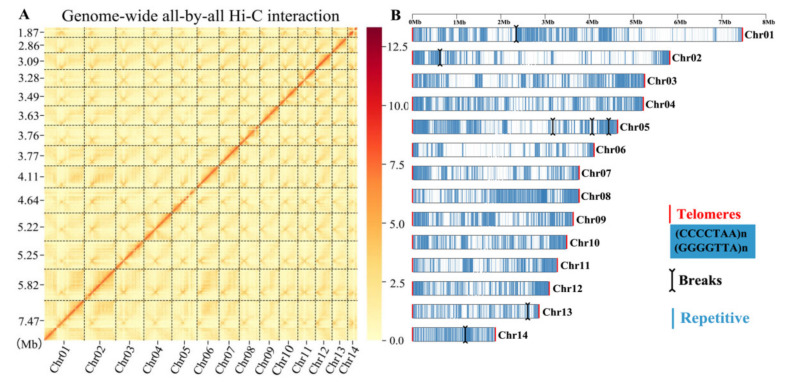
Hi-C interaction heatmap of all bins and chromosome assembly. (**A**) Hi-C interaction heatmap of all bins. The numbers on the left represent the chromosome length. (**B**) Chromosome structures, including the telomeres (red line), breaks (black line), and repetitive sequence (blue line) distributions.

**Figure 3 ijms-23-10484-f003:**
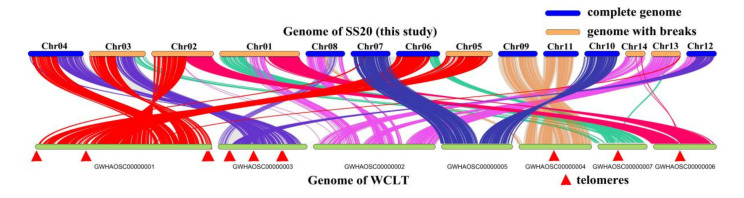
Collinearity between *Wolfipoira hoelen* SS20 and WCLT [24].

**Figure 4 ijms-23-10484-f004:**
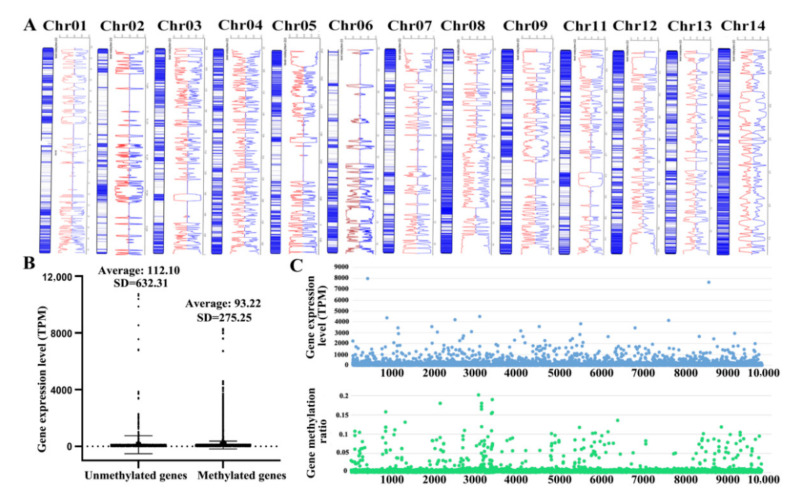
Methylation of transposon and genes, and the relationship of the methylation ratio and gene expression level. (**A**) Transposon location and methylation; (**B**) Gene expression level of methylated and unmethylated genes; (**C**) Methylation ratio and gene expression level.

**Figure 5 ijms-23-10484-f005:**
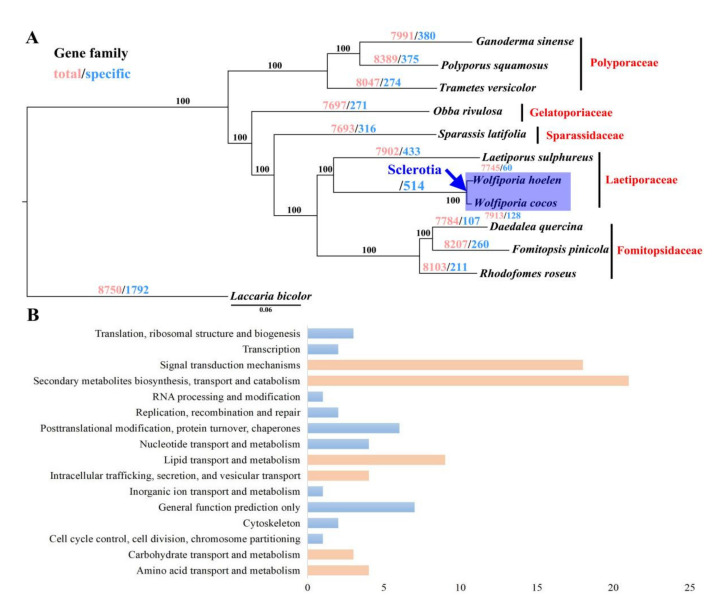
Phylogenetic analysis of *Wolfiporia hoelen* and related species inferred from ML of single-copy orthologous genes (**A**) and function of common expanded gene family in *Wolfiporia hoelen* and *Wolfiporia cocos* annotated with KOG (**B**). Note: The protein mutation rates compared with *W. hoelen* are: *W. cocos* 0.96%, *L. sulphureus* 17.53%, *D. quercina* 17.99%, *R. roseus* 18.01%, *S. latifolia* 19.58%, *F. pinicola* 18.26%, *O rivulosa* 20.57%, *p. squamosus* 21.88%, *T. versicolor* 22.21%, *G. sinensis* 22.81%, and *L. bicolor* 28.11%.

**Figure 6 ijms-23-10484-f006:**
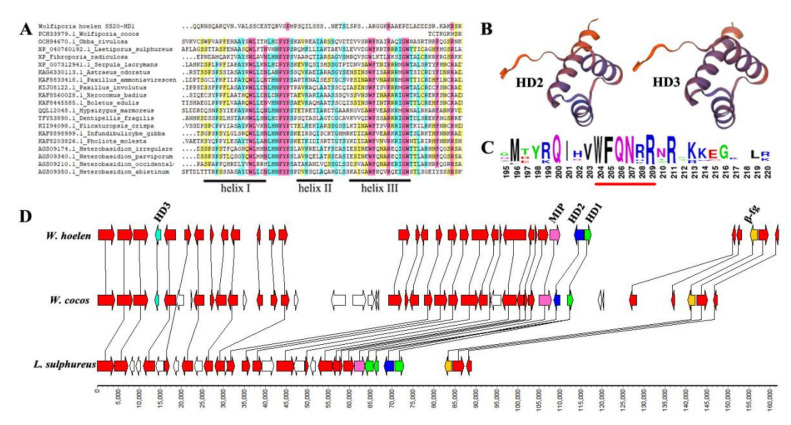
Structure analysis of HD proteins and collinearity of *MAT-A* and surrounding genes among *W. hoelen*, *W. cocos*, and *L. sulphureus.* (**A**) Multiple sequence alignment of HD1 proteins and conserved helix structures; (**B**) Three-dimensional structure of the conserved domain of HD proteins; (**C**) Conserved specific motif of HD2; (**D**) Collinearity of *MAT-A* and its surrounding genes among *W. hoelen*, *W. cocos*, and *L. sulphureus*.

**Figure 7 ijms-23-10484-f007:**
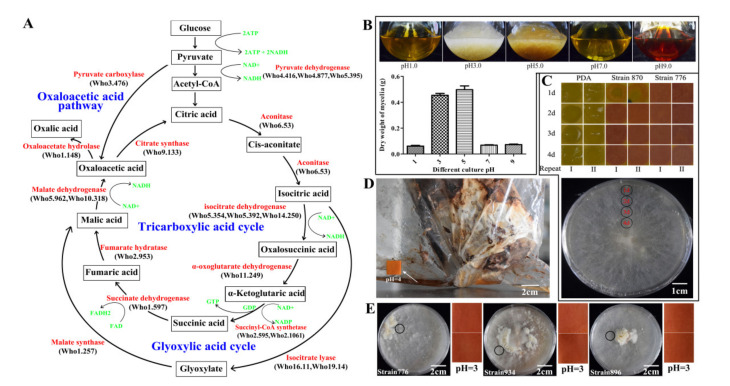
Oxalic acid synthetic pathway in *Wolfiporia hoelen* and the relationship of pH and the physiological process. (**A**) Oxalic acid synthetic pathway in *W. hoelen*; (**B**) Mycelial growth condition and biomass under different pHs under liquid fermentation; (**C**) pH of PDA media on different days after mycelial growth; (**D**) pH of secretion in sclerotia formation; (**E**) pH of media in fruiting. The circles indicate the sites where the samples were taken and pHs were measured.

**Figure 8 ijms-23-10484-f008:**
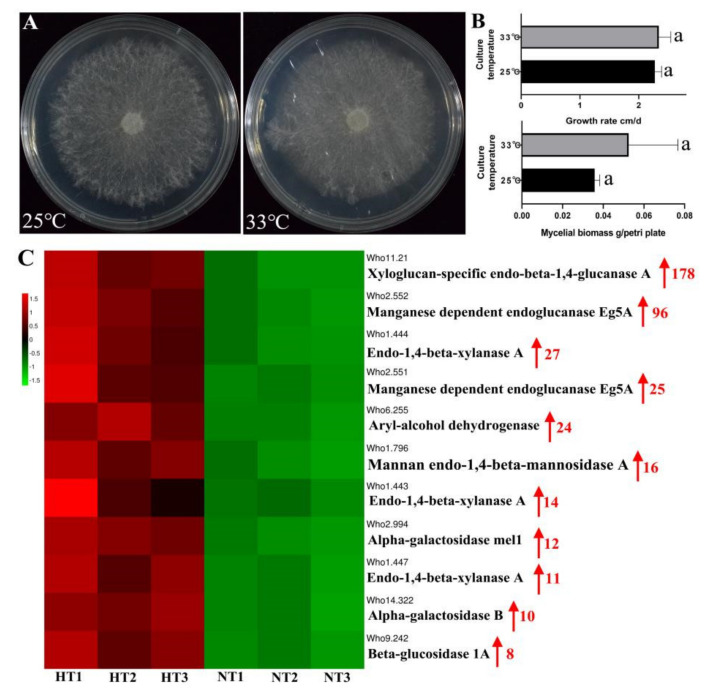
Transcriptome sequencing samples, biological characters, and differential gene expression. (**A**) Colony morphology of CGMCC 5.545 cultured under 25 °C and 33 °C; (**B**) Growth rates and mycelial biomass of CGMCC 5.545 cultured under 25 °C and 33 °C; (**C**) Differential express genes of significantly up-regulated genes. Note: NT presents mycelia cultured under normal temperature, 25 °C, and HT presents mycelia cultured under highest tolerance temperature, 33 °C. The a in subfigure B indicate the significant difference at *p* = 0.05 by Duncan’s multiple range tests. Red arrow in subfigure C present up-regulation expression and the number indicated upregulated multiple.

**Figure 9 ijms-23-10484-f009:**
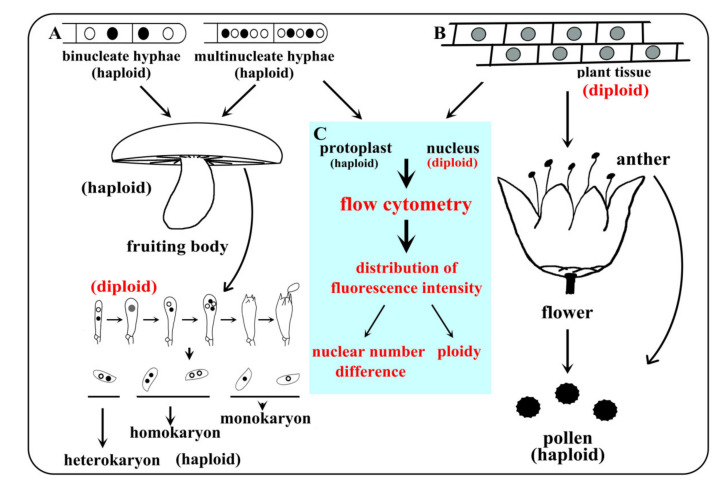
Ploidy change during developmental processes and ploidy analysis routes through flow cytometry in basidiomycetes and plants. (**A**) Ploidy of basidiomycetes at different developmental stages, with the diploid nucleus formed after karyogamy only in basidia; (**B**) Ploidy of plant at different developmental stages, with the haploid nucleus formed after meiosis only in pollen; (**C**) Ploidy analysis for plants and nuclear number analysis for basidiomycetes by flow cytometry.

**Figure 10 ijms-23-10484-f010:**
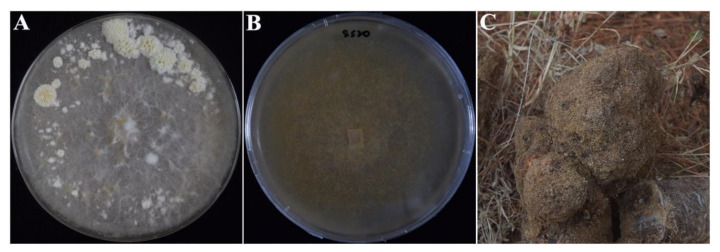
Fruiting bodies, sclerotia, and homokaryotic strain SS20 of *W. hoelen* strain CGMCC 5.545. (**A**) Fruiting bodies of *W. hoelen* strain CGMCC 5.545; (**B**) Homokaryotic strain SS20; (**C**) Artificially cultivated sclerotia of *W. hoelen* strain CGMCC 5.545.

**Table 1 ijms-23-10484-t001:** Classification of the repeat sequences in the genome of *Wolfiporia hoelen*.

Class	Order	Super Family	Number of Elements	Length of Sequence (bp)	Percentage of Sequence (%)
Class I			39,473	23,696,976	36.76
	LTR		35,804	22,399,747	34.74
		Gypsy	18,134	16,432,277	25.49
		Unknown	16,685	5,739,423	8.9
		Copia	896	168,232	0.26
		Other	89	59,815	0.09
	LINE		3189	1,222,889	1.9
		Unknown	2902	1,053,895	1.63
		Tad1	258	163,271	0.25
		Other	29	5723	0.01
	SINE		480	74,340	0.12
		Unknown	480	74,340	0.12
Class II			14,698	6,127,627	9.5
	DNA		6000	3,580,746	5.55
		MULE-MuDR	175	156,180	0.24
		Unknown	5286	3,106,855	4.82
		TcMar-Sagan	274	121,948	0.19
		Other	265	195,763	0.3
	MITE		8620	2,398,308	3.72
		Unknown	8620	2,398,308	3.72
	RC		78	148,573	0.23
		Helitron	78	148,573	0.23
Total TEs			54,171	29,824,603	46.26
Tandem Repeats			3964	152,603	0.24
	tandem_repeat		2380	133,712	0.21
	SSR		1584	18,891	0.03
Unknown			4778	1,324,782	2.05
Simple repeats			82	7498	0.01
Other			5	274	0
Low complexity			1	221	0
Total Repeats			63,001	31,309,981	48.56

**Table 2 ijms-23-10484-t002:** Genome characteristics of strains of *W. hoelen*.

	CGMCC 5.78	WCLT	SS20
Sequencing strategy	HiSeq 2000 Illumina and a fosmid-to fosmid strategy	HiSeq2500 Illumina and SMRT technology on the PacBio	Novaseq6000 Illumina and SMRT technology on the PacBio
Genome size (Mb)	50.6	62	64.44
Number of scaffolds	351	145	78
N50 of scaffolds (kb)	835	1599.1	3760
Anchored to chromosome (Mb)		61.127	58.26
Number of protein-coding genes	10,908	11,906	10,567
Average gene length (bp)	1829	1332.76	2004
Percentage of repeat sequences (%)	-	46.6	48.56
Transposable elements (%)	33.5	-	46.26
GC content (%)	51.7	51.86	50.15
Reference	[23]	[24]	This study

**Table 3 ijms-23-10484-t003:** Protein collinear ratio of different chromosomes comparing *Wolfipoira hoelen*, *Wolfiporia cocos*, and *Laetiporus sulphureus*.

*Wolfiporia hoelen*	Chr01	Chr02	Chr03	Chr04	Chr05	Chr06	Chr07
*Wolfiporia cocos*	76.88%	82.98%	73.80%	76.48%	79.31%	81.90%	73.20%
*Laetiporus sulphureus*	63.38%	70.10%	56.19%	51.03%	57.79%	71.64%	56.80%
** *Wolfiporia hoelen* **	**Chr08**	**Chr09**	**Chr10**	**Chr11**	**Chr12**	**Chr13**	**Chr14**
*Wolfipoira cocos*	67.26%	68.89%	69.84%	77.04%	72.06%	75.51%	29.34%
*Laetiporus sulphureus*	39.64%	46.95%	48.79%	38.90%	35.03%	39.33%	0%

## Data Availability

Raw sequences of both PacBio long-read sequencing and Illumina short-read sequencing were submitted to NCBI SRA (http://www.ncbi.nlm.nih.gov/sra (accessed on 21 March 2022)) under BioProject PRJNA644235. Transcriptome data were submitted under BioProject PRJNA860791.

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
