# Peer review of "Homokaryotic High-Quality Genome Assembly of Medicinal Fungi *Wolfiporia hoelen* Reveals Auto-Regulation and High-Temperature Adaption of Probable Two-Speed Genome"

_ijms, 2022, doi:10.3390/ijms231810484_

Round 1

Reviewer 1 Report

Wolfiporia hoelen is an important medicinal fungus. The manuscript authored by Li et al. reported that the homonuclear genome of Poria cocos was sequenced with high quality to analyze the genetic characteristics of W. hoelen. By analyzing the genomic sequence, methylation level and transcriptome of W. hoelen, the genetic background of W. hoelen, such as genome homeostasis, high temperature adaptation strategy, sclerotic development and mating gene, was analyzed. This work provides detailed genetic background information for the systematic understanding of the biological regulation mechanism in W. hoelen, and the two speed genome, cytosine methylation and other viewpoints mentioned in the paper are very interesting. However, the following problems need to be solved before the article is published.

 Major comment

Although the author discussed the two speed genome in the discussion part, in the results part the data support for the existence of the two speed genome in W. hoelen needs further clarification.

There are similar problems with the auto regulation mentioned by the author, which needs further argumentation, and the support of result to the viewpoint should be clarified.

 Minor comment

It is suggested to further improve the information in the notes of all drawings to ensure that the reader can understand those Figures.

Figure 6 is missing the annotation of D.

Figure 7a is not quoted in the manuscript.

Figure 7b, the photo is too dark to see the mycelium clearly.

Figure 8, HT and NT shall be indicated in full names in the notes.

Author Response

Major comment

Although the author discussed the two speed genome in the discussion part, in the results part the data support for the existence of the two speed genome in W. hoelen needs further clarification.

R: Thanks for your suggestion, the support evidences have been added in lines: 242-245.

There are similar problems with the auto regulation mentioned by the author, which needs further argumentation, and the support of result to the viewpoint should be clarified.

R: Thanks for your suggestion, added in lines: 272-276.

Minor comment

It is suggested to further improve the information in the notes of all drawings to ensure that the reader can understand those Figures.

R: Thanks for your suggestion, we have further revised the notes of all the Figures.

Figure 6 is missing the annotation of D.

R: Thanks for your careful observation, the annotation of figure 6D has been added in lines: 347-348.

Figure 7a is not quoted in the manuscript.

R: Added in line 399.

Figure 7b, the photo is too dark to see the mycelium clearly.

R: The lightness of figure 7b was improved to clearly show the mycelia.

Figure 8, HT and NT shall be indicated in full names in the notes.

R: The note has been added to explain NT and HT.

Reviewer 2 Report

In the manuscript entitled “Homokaryotic high-quality genome assembly of medicinal fungi Wolfiporia hoelen reveals auto-regulation and high- temperature adaptive of probable two-speed genome”, the authors used PacBio sequencing and Hi-C scaffolding technology to assemble the homokaryotic genome of Wolfiporia hoelen. In China and other East Asian countries, the sclerotia from W. hoelen is extensively used for medical purposes. The benefits of W. hoelen sclerotia have long been recognized in China where its cultivation goes back 1500 years. This study is the first to assemble the genome of high-quality homokaryotic W. hoelen. The authors confirmed that the strain used for the genome sequencing was indeed homokaryotic. Since W. hoelen is cultivated in fields, it must endure oppressively high temperatures during the summer season. However, the adaptive mechanism that allows this plant to grow in elevated temperature has been elusive. In this study, the authors analyzed the high temperature adaptation of this plant, in addition to the genome structure, transposons, methylation and mating genes. Their analysis revealed that W. hoelen may have a two-speed genome and genome stability is conserved due to the uniformity between methylation and transposon. Furthermore, their analysis suggests that the high-temperature adaption of W. hoelen may be due to the increased expression of specific decomposition enzymes, ROS clearance genes, unsaturated fatty acids biosynthesis, etc. This study provides significant new information about W. hoelen genome stability and structure and also offers a wealth of new data for other scientists to explore in their work on specific W. hoelen genes.

The manuscript is well-written and the analysis done on the genome is quite detailed. I could not detect any errors with the experiments or analysis present by the authors. The analysis and experiments support the conclusions made.

Author Response

Thanks for your kind evaluation!

Reviewer 3 Report

·         The bioactive compounds from Wolfiporia hoelen with the potent activities of Wolfiporia hoelen in addition to its growing season should be included in the introduction.

·         Line 55: explain the differentiation between American and Chinese W. hoelen. Is this differentiation in whole genome or in partial genomes or in morphological characters?

·          Line 103: Specify which partial genome is included in the detected genome.

·         In figure 1, kindly mention the main cluster of Wolfiporia hoelen which is responsible for secondary metabolism , and give examples.

·         In table 1, kindly write the class individual repetition .

·         In phylogenetic analysis section, kindly explain the reason for using different family. Why not different genus from the same family?

·          In phylogenetic analysis section, kindly mention the mutation rate between different species.

·         In figure 6, part A, there are many gabs in the sequence, please give the explanation.

·         In figure 6, part A, kindly where is the source of energy, as well show the number of ATPs and ADPs in this cycle.

·         In figure 6, explain the effect of pH on the tricarboxylic cycle.

·         In material and methods, kindly mention the place and date of sample collection.

Author Response

Thank for your kind suggestions.
